# Multi-Scale Spatiotemporal Pattern Analysis and Simulation (MSPAS) Model with Driving Factors for Land Cover Change and Sustainable Development Goals: A Case Study of Nepal

**Wenqi Jia [1,2], Xingfa Gu [1,3,4], Xiaofei Mi [1,5,\*], Jian Yang [1], Wenqian Zang [1,5], Peizhuo Liu [1,6], Jian Yan [1], Hongbo Zhu [4], Xuming Zhang [1,7] and Zhouwei Zhang [1]**

1. Aerospace Information Research Institute, Chinese Academy of Sciences, Beijing 100094, China
2. College of Urban and Environmental Sciences, Central China Normal University, Wuhan 430079, China
3. University of Chinese Academy of Sciences, Beijing 100049, China
4. School of Remote Sensing and Information Engineering, North China Institute of Aerospace Engineering, Langfang 065000, China
5. Langfang Research and Development Center for Spatial Information Technology, Langfang 065001, China
6. School of Instrumentation and Optoelectronic Engineering, Beihang University, Beijing 100191, China
7. International Institute for Earth System Science, Nanjing University, Nanjing 210023, China
* Correspondence: mixf@aircas.ac.cn

**Abstract:** In pursuit of Sustainable Development Goals (SDGs), land cover change (LCC) has been utilized to explore different dynamic processes such as farmland abandonment and urban expansion. The study proposed a multi-scale spatiotemporal pattern analysis and simulation (MSPAS) model with driving factors for SDGs. With population information from the census, multi-scale analysis criteria were designed using the combination of administrative and regional divisions, i.e., district, province, nation and ecological region. Contribution and correlation of LCC or population were quantified between multiple scales. Different kinds of driving factors were explored in the pattern analysis and then utilized for the definition of adaptive land suitability rules using the Cellular Automata-Markov (CA-Markov) simulation. As a case study of the MSPAS model, Nepal entered into a new era by the establishment of a Federal Republic in 2015. The model focused on four specific land cover classes of urban, farmland, forest and grassland to explore the pattern of Nepal's LCC from 2016 to 2019. The result demonstrated the performance of the MSPAS model. The spatiotemporal pattern had consistency, and characteristics between multiple scales and population were related to LCC. Urban area nearly doubled while farmland decreased by 3% in these years. Urban areas expanded at the expense of farmland, especially in Kathmandu and some districts of the Terai region, which tended to occur on flat areas near the existing urban centers or along the roads. Farmland abandonment was relatively intense with scattered abandoned areas widely distributed in the Hill region under conditions of steep topography and sparse population. The MSPAS model can provide references for the development of sustainable urbanization and agriculture in SDGs.

**Keywords:** MSPAS; SDGs; spatiotemporal pattern; multi-scale analysis; land suitability with driving factors; CA-Markov; urban expansion; farmland abandonment

## 1. Introduction

LCC is widely studied in the social and environmental phenomena [1,2]. In recent years, great LCC has occurred around the world due to population explosion, climate change and other problems, especially in some developing countries. It also brings some consequences [3], such as natural disasters [4], carbon emissions [5], air pollution [6], water degradation [7] and so on. Nowadays, sustainable development has been accepted by countries worldwide [8,9]. The 2030 Agenda for Sustainable Development includes 17 SDGs and 169 targets for the period 2016–2030. Among these goals, SDG 2 [10] focuses

on sustainable agriculture, which aims to end hunger, achieve food security, double the agricultural productivity and increase incomes of small-scale food producers. SDG 11 [11] proposes to enhance inclusive, sustainable and safe urbanization and reduce the adverse environmental impact of urban areas. The above goals are also highly relevant and essential to other SDGs, including SDG 8 for economic growth, SDG 9 for infrastructure and SDG 15 for life on land. Asia and Africa have been hot spots of SDGs because hunger and poverty, effects of climate change or extreme events, growth of population and urbanization in these areas are greatly higher than the rest of the world [12]. The urban growth has been multiple times greater than previous estimates of the worldwide cities with an unprecedented global urbanization [13].

Remote sensing data has been widely used in geographic simulation [14,15], LCC monitoring [16–19] and SDGs, with advantages of large area coverage and short revisit time. It can provide rich temporal, spatial and spectral information for different land covers [20–22]. Several methods have been developed to simulate LCC quantitatively and spatially. As a stochastic model, the Markov chain calculates transition probabilities between various land cover classes based on data of two times [23,24] and is well fit for prediction over a short time period. However, it only estimates the magnitude of transition and neglects spatial information. A cellular automaton is a bottom-up and discrete model with the ability to simulate many complex processes [25–27], which considers the effect of neighborhood spatially. Based on CA, SLEUTH uses slope, land use, exclusion, urban extent, transportation and hill shade as inputs to monitor urban change [28]. CA-Markov combines the advantages of CA and Markov chain models, which is much suitable for the simulation of spatiotemporal change [29,30]. It can also integrate different driving factors simultaneously. For SDGs, remote sensing data are commonly utilized for generating indicators and evaluating the achievement of targets in SDGs [31], including multi-resolution indicators, environmental indicators and integrated indicators [32].

There are three points of significance and necessity in this work. (1) Most methods analyze the LCC spatiotemporal pattern at a single scale of district [33], province [34], region [35] or nation [36], which neglect that these scales are related to each other and act together to form the pattern of a country. The significance of multi-scale analysis is as follows. The policy design and making are always based on administrative or regional divisions, which need guidance and emphasis according to the situation in different places. For a country, it should be clear which province needs to be highly concerned due to its worse situation of ecological or developmental problems than other provinces. The diversities of all districts in a province also influence urban planning and function. As references for policymaking and allocation of resources or government spending, studies on multi-scale LCC can reflect the intensity of ecological and socio-economic processes or problems in different provinces, districts and regions with the demand of protection and governance. Farmland to urban is related to urban planning and farmland to forest corresponds to arable land protection. Land management is always divided into multiple levels and conducted in nations, provinces or districts. How to establish and quantify the connection of LCC between different scales is a question worth discussing. (2) For driving factors, the administrative scale is closely related to political and socio-economic factors such as population, transport and economy [37], while the regional scale is relevant to ecology and geography. Driving factors lead to the similarities and diversities of LCC at multiple scales, and various land cover types also have different sensitivities to the same driving factor. There are large diversities of driving factors for land covers in different areas. For instance, LCC of urban areas is sensitive to population and transport, while LCC of forests mainly depends on natural factors. It is necessary to analyze the impact of driving factors on specific land cover and spatiotemporal patterns [38,39], and then utilize the analysis result for a better simulation. The population can significantly influence LCC by human activities, especially for urban [40] and farmland areas, with a high speed of variation in a short time [41]. Many countries have attached great importance to the population factor and conducted a census. Quantifying the relationship between LCC and

population can better explore the interaction between the human and ecology systems. (3) In regard to SDGs, many researchers study them from the perspective of land change, land suitability, land management and efficiency [12], which are highly related to food security and production, urban planning, land systems and other issues about SDGs. Land cover data can provide multi-scale spatial information and multi-temporal monitoring for the assessment of SDGs [32]. The calculation and generation of many targets and indicators of SDGs rely on the area of land cover and the population, and dynamic sustainable development is further influenced by LCC. Several related issues have drawn wide attention. Firstly, many farmlands are abandoned [42] and gradually turn into forest or grassland, corresponding to conversion from farmland to forest or grassland and food security of SDG 2. At the same time, urban expansion has risen rapidly at the expense of farmland, corresponding to conversion from farmland to urban area and the sustainable urbanization of SDG 11 [43,44]. Understanding LCC in urban area and farmland is crucial to supporting the SDGs for a specific country.

Therefore, the study proposed a multi-scale spatiotemporal pattern analysis and simulation model, named the MSPAS model, with driving factors. The model was divided into an analysis part and a simulation part. Corresponding to the above three points of significance and necessity, the innovations of the model are as follows: (1) With population information from the census, multi-scale analysis criteria for LCC spatiotemporal patterns were designed based on administrative and regional divisions, i.e., district, province, nation and ecological regions. The contribution and correlation of LCC or population between different scales were calculated for the quantification of the multi-scale connection. The spatiotemporal pattern was analyzed quantitatively and spatially with driving factors. (2) The impacts of different driving factors on specific land cover were explored in the analysis part and then utilized for the definition of a novel adaptive land suitability rule in the CA-Markov simulation part. (3) The MSPAS model was applied to SDGs by combining the LCC pattern with factors of policies and planning. The MSPAS model for LCC is based on the administrative and regional divisions, which can be utilized in many countries with international applicability.

## 2. Study Area and Materials

### 2.1. Study Area

As a landlocked country between China and India, Nepal was affirmed as a federal democratic republic (FDR) in 2015 and changed dramatically in its administration and development pattern. The GDP and total population of Nepal have achieved a pretty steep growth from 2016 to 2019, which were much faster than the past years, and can be references for supposing that land cover of Nepal has a potential to change greatly in a different way. The research took Nepal as a case study for the MSPAS model. Located in South Asia, the country is divided into three ecological regions as shown in Figure 1. In the north of Nepal, the Mountain region constitutes the central part of the Himalayan range. Covering 68% of the whole area, the Hill region is situated in the middle of Nepal with an altitude between 1000 m and 4000 m. With flat terrain, fertile soil and suitable climate, the Terai region has an altitude between 67 m and 300 m above sea level and provides the largest share of agriculture production among three regions, accommodating 50.3% of the national population in 2011.

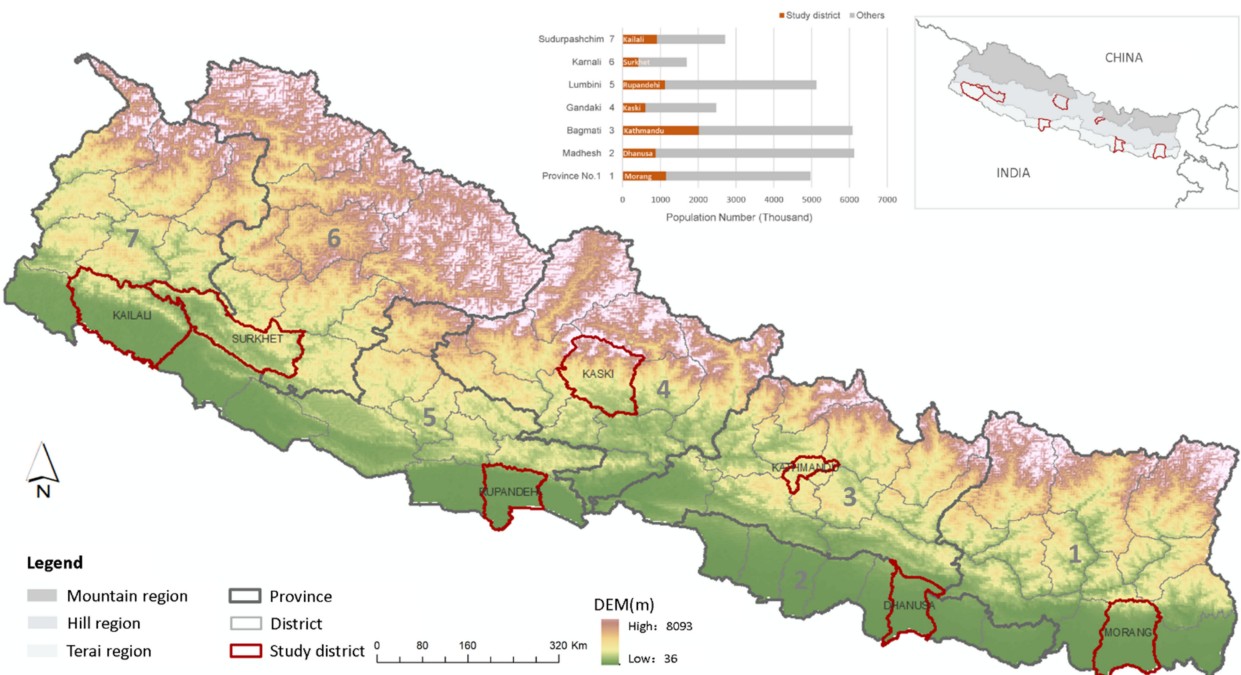

**Figure 1.** Study area and population number of seven provinces and districts from the census in 2021. The boundary information is from the official boundary of Nepal.

For multi-scale administrative divisions, Nepal was divided into 7 provinces, 77 districts and 753 local units. In each of the 7 provinces, the most populous district was chosen as an interest district based on population information from census, i.e., Morang in Province No.1, Dhanusa in Madhesh, Kathmandu in Bagmati, Kaski in Gandaki, Rupandehi in Lumbini, Surkhet in Karnali and Kailali in Sudurpashchim. For ecological divisions, Kailali, Rupandehi, Morang and Dhanusa are located in the Terai region, and Kathmandu, Surkhet and Kaski are in the Hill region. Located in different provinces and regions, the interest districts can be utilized for the analysis of multi-scale change patterns with certain representativeness. The environmental condition, population number and economic development of the seven districts vary dramatically. Most districts in the Terai region have larger populations than ones in the Hill region. Kaski is near the Mountain region and covers several large mountains. As Nepal's capital, Kathmandu has the highest population, while Surkhet has the lowest one in the seven districts.

*2.2. Data*

In the study, two kinds of data were utilized, as shown in Table 1. One is a land cover dataset from remote sensing images, the other one is data of driving factors. Land cover data from 2016 to 2019 were gathered from The International Centre for Integrated Mountain Development (ICIMOD), which are classified into nine different classes: urban, farmland, grassland, forest, water body, snow and glacier, riverbed, bare soil and bare rock. Snow and glacier, bare soil and bare rock are mostly in the Mountain region but rarely exist in the seven study districts. Water body and riverbed influence the distribution of urban area and farmland, but change slightly in a short period of time. As shown in Figure 2, the major land cover classes in seven districts consist of urban area, farmland, forest and grassland. Therefore, this study analyzed change and interaction between the major four land cover classes by using the data of nine classes from ICIMOD. Driving factors consist of slope, elevation, road, river, population and policies. Topography data was obtained from ASTER GDEMV2 and OpenStreetMap provided the information on rivers and roads. Population distribution data in 2016 was mapped by WorldPop with a resolution of 100 m. All data covered the same area with the same resolution of 30 m after image resampling. Statistical population number was counted by the government

of Nepal at district, provincial and national scales, and GDP data was gathered from The World Bank. Political factors from the government of Nepal were relevant to SDGs and the development of urban or agricultural areas.

**Table 1.** Data utilized in the study.

| Category | Data | Source | Property |
|---|---|---|---|
| Physical | Land cover maps | ICIMOD | Land cover classification |
| | Topography | ASTER GDEMV2 | Elevation and slope |
| | River | OpenStreetMap | Accessibility to water resource |
| Socio-economic | Road | OpenStreetMap | Accessibility to transport |
| | Population distribution | WorldPop | Spatial distribution of population |
| | Population number | Central Bureau of Statistics (CBS), Government of Nepal | District, provincial and national population |
| | GDP | World Bank | Economy |
| Political | Policy and planning | Government of Nepal | Political information about SDGs, agricultural and urban areas |

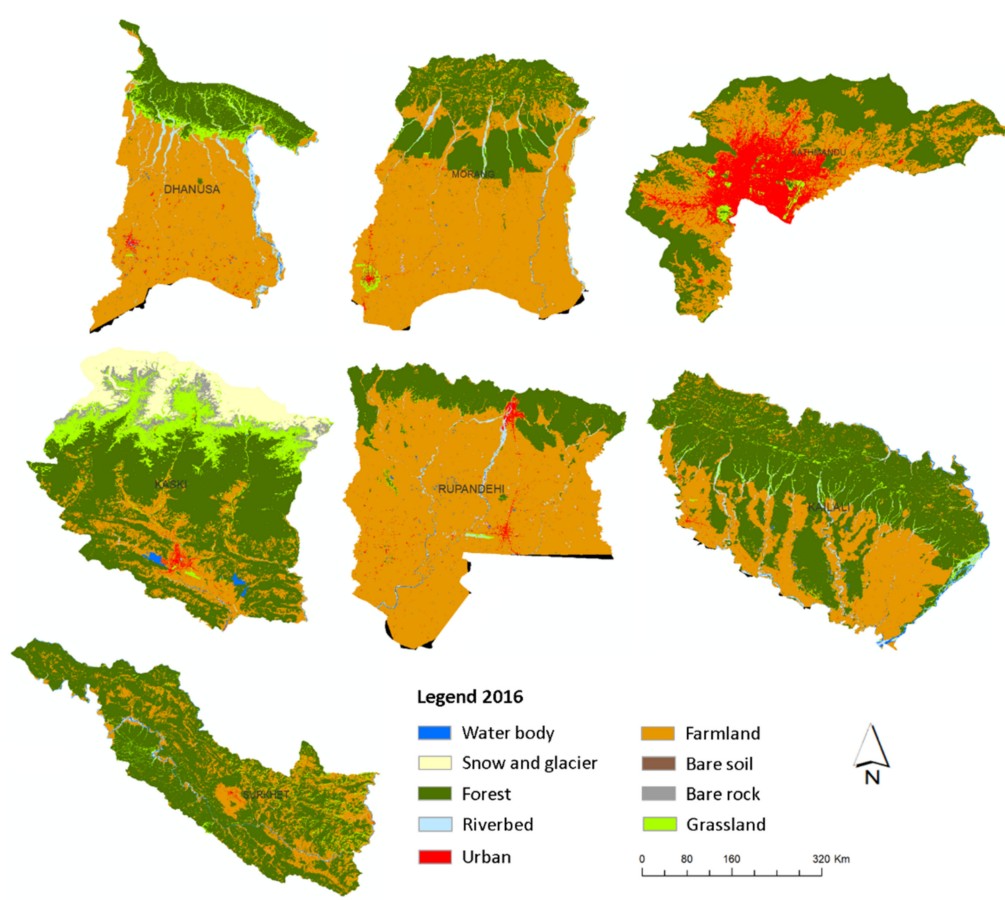

**Figure 2.** Land cover of the seven interest districts in 2016.

## 3. MSPAS Model with Driving Factors

### 3.1. Multi-Scale Analysis Criteria for the LCC Spatiotemporal Pattern

As shown in Figure 3, multi-scale analysis criteria are designed with the population factor to explore the patterns of LCC between different spatial scales. GDP and population information provide the preliminary references for study time and area. Multi-scale analysis

is based on administrative division and regional division. Administrative division can be divided into nation, province and district, while regional division can utilize ecological or geographical zones. The most populous district in each of the provinces is selected as an interest district. The ecological region where the majority of interest districts fall is regarded as the corresponding region of this district, and then interest districts can be reclassified into several categories by the ecological regions they belong to. This new classification synthesizes both socio-economic and physical factors from the above administrative and regional divisions, which becomes the base of pattern analysis at the district scale. After overlaying different divisions onto the LCC map and population data, the areas of various kinds of LCC and population number at the scale of nation, province, region and district are obtained. Firstly, the spatiotemporal pattern at a single scale such as interest district or nation is explored spatially and quantitatively to find the characteristics and consistency, respectively. Secondly, the contributions of LCC or population are calculated to quantify the connection between multiple scales in Equations (1) and (2). There are four situations of contribution between multiple scales being explored in the study: interest district's contribution to its corresponding province (DP), combined districts' contribution to nation (CN), province's contribution to nation (PN) and region's contribution to nation (RN). Multi-scale contributions can reflect the importance of components in the first smaller scale to the whole part in the second larger scale for specific land change or population. The combined districts represent the sum of all interest districts in a country. Thirdly, in Equation (3), the correlations between multi-scale contributions are calculated to quantify the relevance between different land change types in the four situations. Population is also considered in the correlation analysis.

$$C_{\mathrm{AB}}^{mn} = \frac{S_{\mathrm{A}}^{mn}}{S_{\mathrm{B}}^{mn}} \times 100\% \tag{1}$$

where A is the first smaller scale and B is the second larger scale, $C_{\mathrm{AB}}^{mn}$ means LCC contribution from class $m$ to class $n$ between scale A and scale B, $S_{\mathrm{A}}^{mn}$ is the area of land change from class $m$ to class $n$ at scale A and $S_{\mathrm{B}}^{mn}$ represents the same meaning at scale B. For instance, scale A is the interest district and scale B is its corresponding province in DP, while scale A is one ecological region and scale B is the whole country in RN. In CN, $S_{\mathrm{A}}^{mn}$ is the sum of land change area in all interest districts and $S_{\mathrm{B}}^{mn}$ is the land change area of the whole country. All the area information is obtained by overlaying the LCC map onto administrative or regional divisions.

$$CP_{\mathrm{AB}} = \frac{P_{\mathrm{A}}}{P_{\mathrm{B}}} \times 100\% \tag{2}$$

where $P_{\mathrm{A}}$ is the population at scale A and $P_{\mathrm{B}}$ is the one at scale B and $CP_{\mathrm{AB}}$ means population contribution between scale A and scale B.

$$r = \frac{\sum (x - \overline{x})(y - \overline{y})}{\sqrt{\sum (x - \overline{x})^2 \sum (y - \overline{y})^2}} \tag{3}$$

where $x$, $y$ are the two types of LCC contributions or population contribution in one specific situation of DP, CN, PN or RN, $\overline{x}$ and $\overline{y}$ are the mean value of variations. $r$ is the correlation between two variations, which reflects the connection between different LCC or population contributions in the situation.

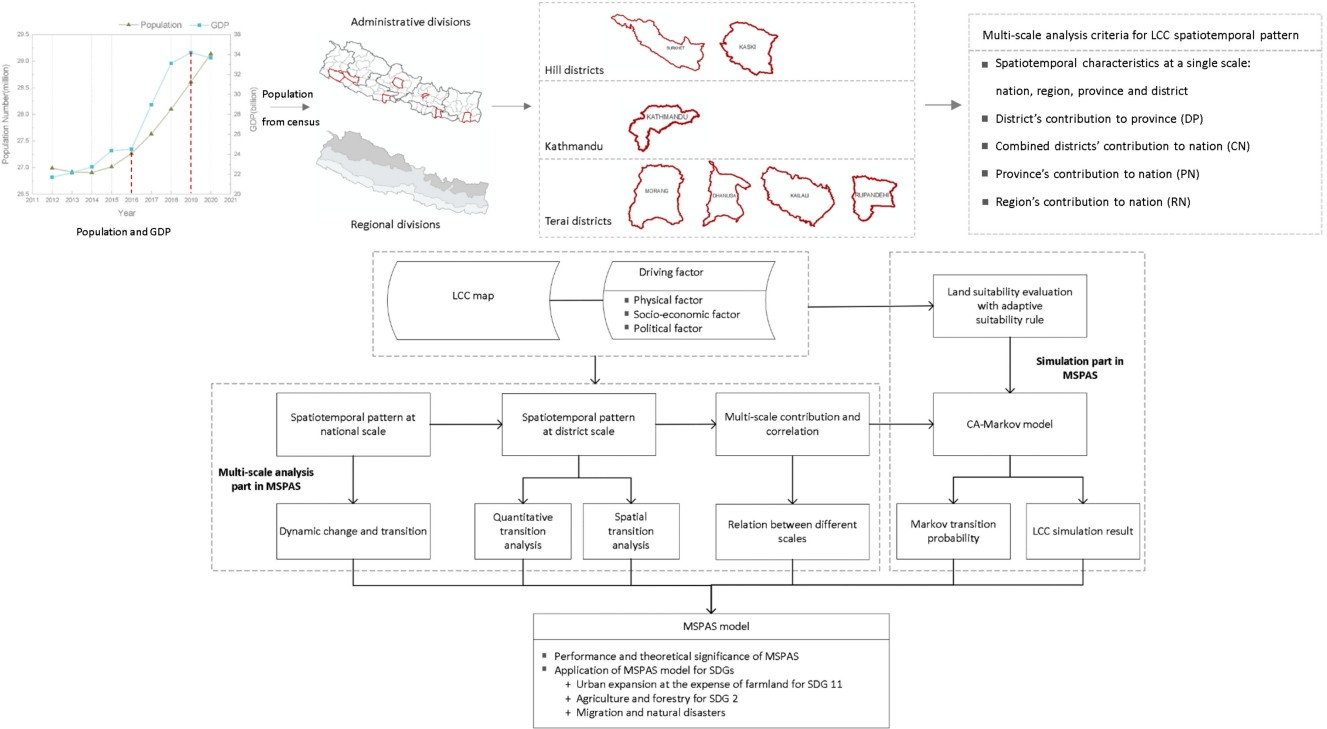

**Figure 3.** Flowchart of MSPAS model.

### 3.2. Driving Factors and Adaptive Land Suitability Evaluation

To understand dynamic LCC more comprehensively, different kinds of driving factors are utilized in the two parts: the spatiotemporal pattern analysis part and the land suitability evaluation of the CA-Markov simulation part. In the analysis part, the interest districts are selected by population factors from the census, while ecological divisions are closely related to slope and elevation. Multiple maps of driving factors are overlaid onto the LCC maps to analyze the characteristics of land change. The spatial relation between driving factors and LCC distribution can reflect the impact of these factors on specific land cover. After obtaining the relationship between LCC and driving factors in the analysis part, important factors for specific land cover are determined and then utilized for adaptive land suitability evaluation in the simulation part. The factors of policies and planning are finally gathered to apply the MSPAS model to land management and SDGs for a country.

The adaptive land suitability rule is proposed to solve large diversities of driving factors for various land covers in different areas, which quantifies the suitability and impact of selected driving factors by the histogram of the actual situation. In multi-criteria analysis, the key steps of developing the land suitability rule are fuzzy evaluation and weight definition for driving factors. The membership functions of fuzzy evaluations include sigmoidal, J-shaped and linear functions, which can describe monotonically increasing, decreasing and symmetric relations, and obtain standardized factor maps by the definition of control points. Then, the weights need to be set to indicate the importance of each factor, and finally, the land suitability for a specific land cover is evaluated. Note that the appropriate ranges of one driving factor for different land covers are not the same, and the situations of driving factors also vary greatly in different areas. The adaptive land suitability rule is proposed to solve the problem. The control points and fuzzy function are adjusted to match the specific actual situation in the target district and vary with the change of land cover class, instead of being fixed and the same in all places. As shown in Figure 4, elevation and slope of farmland or urban areas in one district are masked and extracted separately, and then statistical histograms are obtained as references for the definition of control points. The function with control points is designed to fit with the histogram as well as possible. For instance, the starting control point of the decreasing function for the

slope factor is designed to be consistent with the highest value of the histogram, which is adaptive to the change of target area and land cover class. Weight definition can utilize the result of a driving factor's impact on LCC from the analysis part. A greater impact corresponds to a higher weight. Land suitability maps are finally obtained as the input of the CA-Markov model.

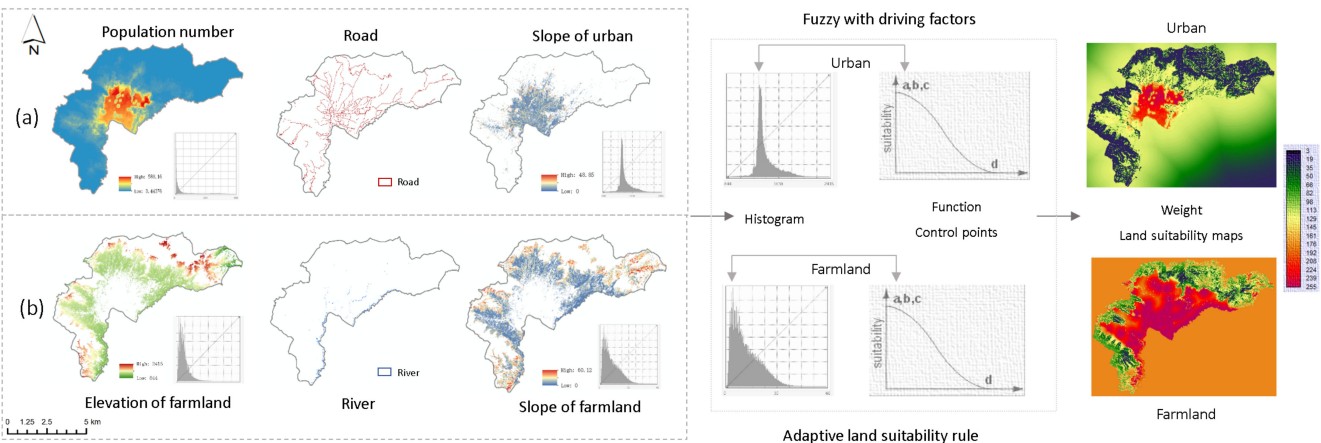

**Figure 4.** Adaptive land suitability rule with driving factors. Kathmandu is taken as an example. (**a**) Driving factors for an urban area. (**b**) Driving factors for farmland. Note that the figure is utilized to describe the adaptive land suitability rule here, and the determination of driving factors for Nepal is introduced in Section 4.4.

### 3.3. Simulation of LCC Using a CA-Markov Method

A CA-Markov method is employed to simulate the spatiotemporal process of LCC based on the combination of CA and Markov chain models [45]. Markov chain utilizes a transition probability matrix between two states to predict land cover change quantitatively, which is very suitable for short term prediction. The next state of time $t + 1$ is only determined by the state of time $t$ with no relation to historical state. However, it neglects the spatial distribution of land cover and is unable to consider the impact of spatial knowledge on LCC. According to Tobler's first law of geography, i.e., "everything is related to everything else, but near things are more related than distant things." [46], the impact of a nearby area on the target area should also be considered. CA is a dynamic and discrete model, which consists of four fundamental parts: cell state, cell space, neighborhood and transition rule [47,48]. Based on transition rule, the next state of a center cell is determined by its initial state and adjacent cells in the neighborhood. Integrated with transition probability, the CA-Markov model takes the effect of neighborhood into account, which can better simulate complex spatiotemporal process. The model is expressed using Equations (4)–(6).

$$\mathbf{P} = \begin{bmatrix} P_{11} & \cdots & P_{1n} \\ \vdots & P_{ij} & \vdots \\ P_{n1} & \cdots & P_{nn} \end{bmatrix} \tag{4}$$

$$C_{t+1} = \mathbf{P} \times C_t \tag{5}$$

$$S_{t+1} = f(S_t, N) \tag{6}$$

where **P** is the transition probability matrix; $P_{ij}$ represents the transition probability from land cover class $i$ to class $j$, $n$ is total number of land cover classes, $C_t$ is the vector of land cover class state at time $t$ and $C_{t+1}$ represents the state at time $t + 1$. In the formula of CA, $f$ represents the transition rule, $N$ is the neighborhood, $S_t$ is the state of center cell at time $t$ and $S_{t+1}$ is the state of one at time $t + 1$.

## 4. Results

Focusing on four land cover classes of urban, farmland, forest and grassland, the model explored four kinds of LCC types in Nepal: forest to farmland, farmland to urban, farmland to forest and farmland to grassland. In the analysis part, spatiotemporal patterns of LCC were firstly analyzed at the single scales of nation and district, respectively, in Sections 4.1 and 4.2. In Section 4.3, the multi-scale contributions of DP, PN, CN and RN and their correlations were calculated to quantify the connection between different scales. The analysis part explored the relationship between driving factors and LCC, then specific factors for the land covers of urban and farmland were determined. In the simulation part of Section 4.4, these factors were employed in the proposed adaptive land suitability rule of the CA-Markov method to simulate the dynamic process.

### 4.1. Spatiotemporal Pattern of LCC at National Scale

The conversions between urban, farmland, forest and grassland are shown in Table 2. Urban area nearly doubled from 2016 to 2019 with a high dynamic degree. Most of the expansion was derived by conversion from farmland to urban areas in each year. Compared to grassland and forest, farmland played a significant role in urban expansion. The change of urban area was not steady. After the rapid increase from 2016 to 2018, change in 2018–19 was very slight. Urban area achieved great growth, but the opposite change from urban to forest, farmland or grassland was very small. Urban expansion corresponded to a decline in farmland area with steady loss rate. Farmland has decreased by 3% over the three years studied. Conversion from farmland to forest and grassland was more stable than from farmland to urban, which contributed about 7‰ to forest area and more than 3‰ to grassland in each year. Forest and grassland were cultivated and changed into farmland, while abandoned farmland turned into forest and grassland gradually. To sum up, farmland was lost quantitatively in the interaction with forest and grassland. Change in the spatial distribution of these land covers was also active annually under frequent mutual conversion. Forest areas expanded and some of them derived from farmland, while grassland reduced every year. The decline of grassland area in 2018–2019 was relevant to the transformation to bare rock beyond the study.

**Table 2.** Dynamic degree of land covers and transition between urban, farmland, forest and grassland from 2016 to 2019 (unit: ‰). In the transition from 2016 to 2019, the number represents the proportion of transition area to total area of the specific land cover in the former year. Dynamic degree reflects the change rate of land area during the years, which is influenced by all other land cover classes.

| 2017 / 2016 | Forest | Urban | Farmland | Grassland | 2018 / 2017 | Forest | Urban | Farmland | Grassland |
|---|---|---|---|---|---|---|---|---|---|
| Forest | 993.12 | 0.74 | 6.25 | 9.82 | Forest | 992.92 | 5.59 | 9.56 | 5.42 |
| Urban | 0.0001 | 990.88 | 0.02 | 0.03 | Urban | 0.0002 | 991.28 | 0.02 | 0.03 |
| Farmland | 7.08 | 144.85 | 980.69 | 3.30 | Farmland | 6.57 | 648.51 | 974.03 | 3.73 |
| Grassland | 3.19 | 15.32 | 1.51 | 890.22 | Grassland | 4.98 | 37.12 | 1.83 | 894.28 |

| 2019 / 2018 | Forest | Urban | Farmland | Grassland | Dynamic degree of land covers (‰) | | | | |
|---|---|---|---|---|---|---|---|---|---|
| Forest | 990.59 | 0.10 | 7.66 | 15.44 | **Year** | **Forest** | **Urban** | **Farmland** | **Grassland** |
| Urban | 0.0008 | 991.92 | 0.04 | 0.04 | 2016–17 | 3.46 | 166.35 | −10.71 | −7.95 |
| Farmland | 7.32 | 0.69 | 980.91 | 4.91 | 2017–18 | 4.68 | 710.55 | −13.74 | −4.06 |
| Grassland | 1.77 | 1.24 | 1.90 | 840.16 | 2018–19 | −0.23 | −3.31 | −8.76 | −78.05 |

### 4.2. Spatiotemporal Pattern of LCC at District Scale

Based on administrative and regional division, the interest districts were selected and classified into three categories: Terai districts, Kathmandu and Hill districts. Morang, Dhanusa, Kailali and Rupandehi were referred to as Terai districts, while Surkhet and Kaski

were categorized as Hill districts in the study. Due to political and historical particularity, the capital Kathmandu in this study was separate from the Hill districts. The spatiotemporal pattern analysis at district scale was based on the above classification. Different kinds of driving factors were combined with LCC to explore their effects on urban and farmland areas, such as population, roads, elevation, slope and rivers.

### 4.2.1. Quantitative Transition Analysis

From Figure 5, urban area expanded while farmland had a loss in all seven interest districts from 2016 to 2019, which was in line with the changing trend at national scale. Kathmandu and the Terai districts had much area of urban expansion. With a reduction in quantity, the change of farmland was opposite that of urban area, due to transition from farmland to urban area and to forest. With the highest original farmland area, Morang lost the largest area of farmland among seven districts. It is notable that the decrease of farmland in the Hill districts was relatively severe compared with the slight increase of urban area. Kathmandu and the Hill districts had a smaller original area of farmland than the Terai districts, but their losses of farmland were huge and even higher than the latter, which means that the decline of farmland was obvious in these districts.

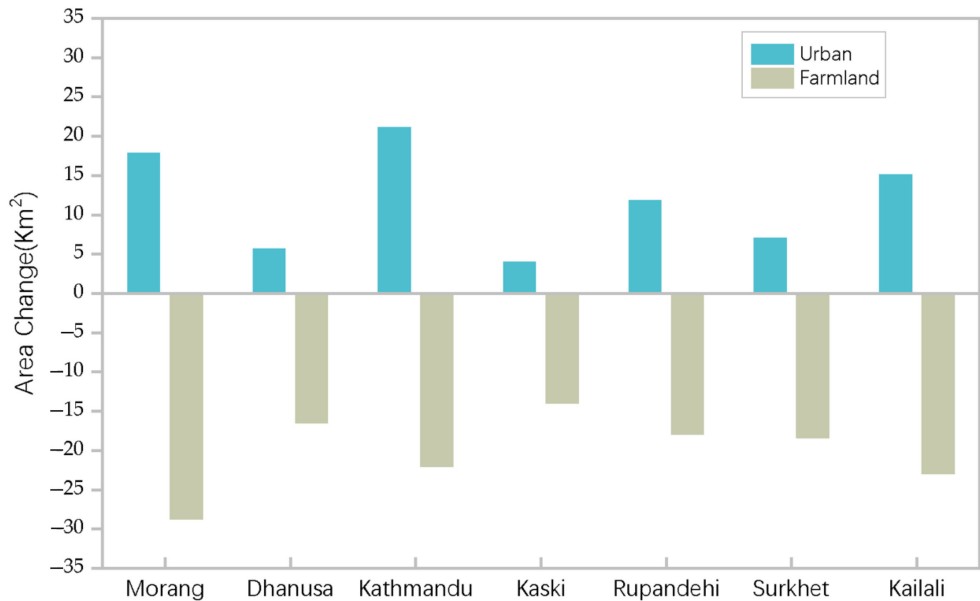

**Figure 5.** Change of urban and farmland area in seven interest districts from 2016 to 2019.

From Figure 6, the internal proportion of farmland to urban areas was the largest in the four kinds of LCC for Kathmandu and the Terai districts, which means that urban expansion at the expense of farmland was obvious in these districts. The Hill districts had the highest area of conversion from farmland to forest in seven districts, even though their original farmland area was the least. Therefore, farmland abandonment was more prominent in the Hill districts. Additionally, a mutual transition between farmland and forest also accounted for a larger internal proportion in the two districts of Surkhet and Kaski than the other districts. From the perspective of internal LCC composition, LCC between urban area and farmland was active in Kathmandu and the Terai districts, while conversion between farmland and forest was relatively intense in the Hill districts. Moreover, the area of farmland to forest was higher than forest to farmland in most districts, and they had correlation with each other in quantity.

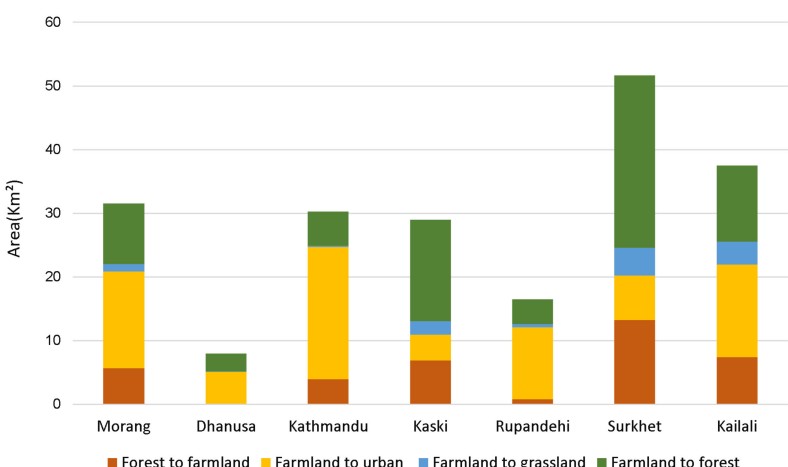

**Figure 6.** The composition of four kinds of land cover changes in seven interest districts from 2016 to 2019.

### 4.2.2. Spatial Transition Analysis

Maps of various driving factors such as elevation, road and river were overlaid onto LCC maps to analyze the characteristics of change patterns and the impact of driving factors spatially. From Figures 2 and 7, it can be seen that spatiotemporal patterns of LCC were different in seven interest districts from 2016 to 2019. Urban expansion was more likely to occur near the existing urban centers. As the hub of Nepal's urbanization, Kathmandu had an obvious urban encroachment on nearby farmland, which mainly happened on the flat area outside the urban core. Some abandoned farmland existed on the surrounding hills and changed into forest. The Terai districts all had vast plain and farmland areas with road networks or rivers across them. Transition from farmland to urban area mostly occurred along the roads, especially in the intersection and convergence parts. A large proportion of urban expansion happened in the southwest Dhanusa, Morang and Kailali with a dense road network. Rupandehi also had a marked increase of urban area near the riverbed and road. Based on the existing built-up areas, there was small-scale urban expansion widely scattered throughout the flat plains in Terai, such as the middle of Morang and Kailali. As for farmland abandonment, it mainly happened on the border of forest and farmland, including the boundary area of vast forests in the north of Dhanusa, Rupandehi and the middle of Morang. One of the special circumstances was the northern area of Morang and Kailali, which was scattered with small farmland in the forest and hill. Unlike large arable land in the southern plain, farmlands in the north of the two districts were relatively smaller than the southern ones, corresponding to more fragmented abandoned areas and active interaction between farmland and forest. Overall, urban expansion was more intense than farmland abandonment in these districts.

With higher elevation, steep slope and smaller population in the Hill region, Surkhet is in the west of Nepal, and Kaski is near the Mountain region. In these Hill districts, conversion from farmland to urban area was concentrated in a limited spatial extent of area near the urban centers with intensive road networks. Farmland of the Hill districts differed from that of the Terai districts in the spatial distribution and size, which was much smaller or presented many fragmented patches on the vast forest or hill [49]. This form corresponded to a higher risk of farmland abandonment with more active conversion from farmland to forest or grassland. The abandoned areas from farmland to forest in the Hill districts also exhibited scattered fragments and were widely distributed in the stretch of large forests or remote mountainous areas. In addition, there were obvious areas converted from forest to farmland around the river of Surkhet. Overall, farmland abandonment in the Hill districts was more widespread and severe than that in Terai.

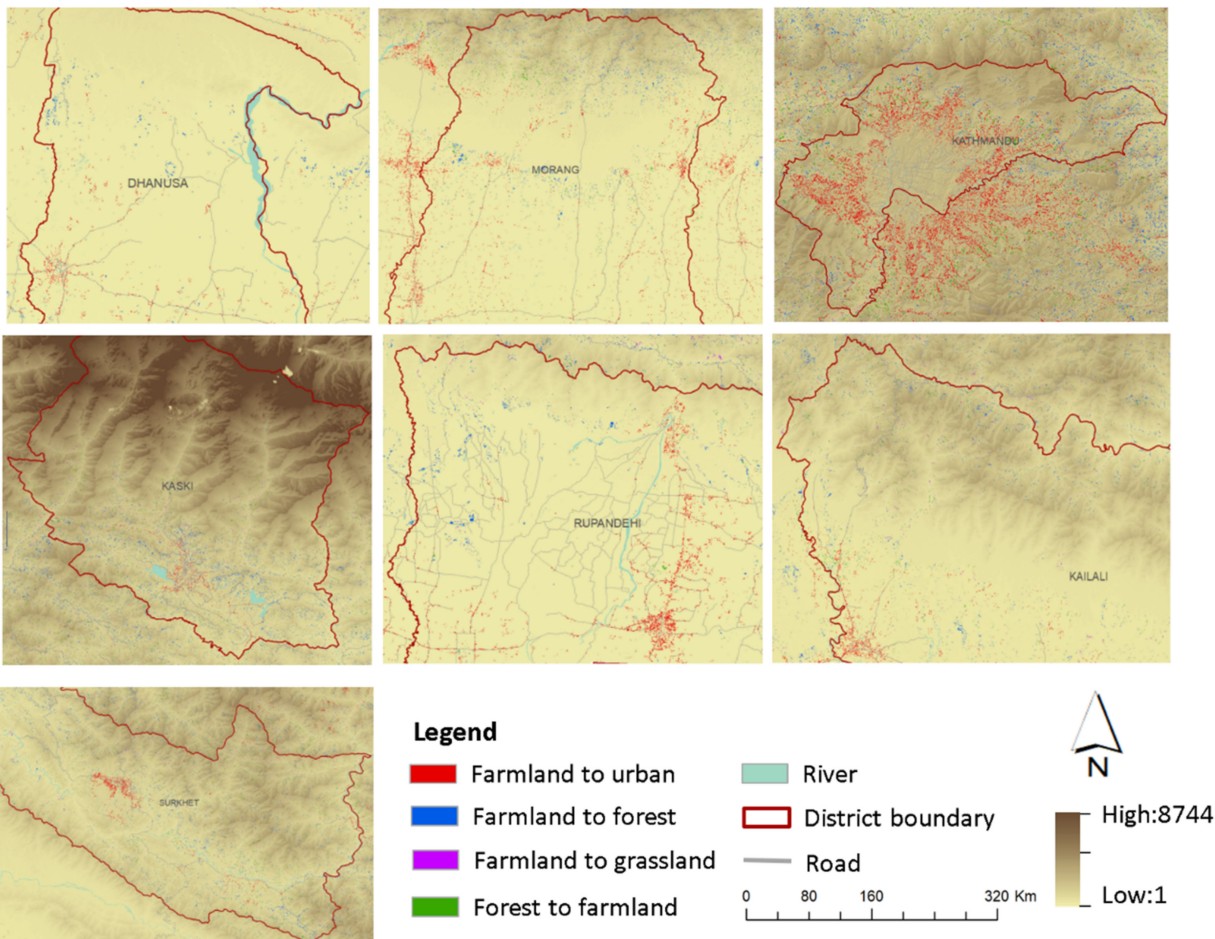

**Figure 7.** Spatial distribution of four kinds of land cover changes from 2016 to 2019. Maps of river, road and topography are overlaid on it. The Terai districts are Morang, Dhanusa, Kailali and Rupandehi. The Hill districts are Surkhet and Kaski.

### *4.3. Multi-Scale Contribution and Correlation*

Multi-scale contribution and correlation were employed for DP, PN, RN and CN. From DP in Figure 8, the contribution of variation from farmland to urban area was relativity larger than the other three variations and closer to the value of population, which implied that contribution of urban expansion was prominent in most interest districts with close relation to population. Note that population contribution was mostly higher than the LCC contribution with the possibility of a population agglomeration effect in the interest district, such as the situation of Kathmandu. Change from farmland to forest in Surkhet had the most contribution to provincial change, corresponding to its problem of farmland loss. From PN, the proportions of four variations were around the value of population, reflecting a better correlation between population and LCC. In seven provinces, Province No.1 took up a large proportion of forest to farmland, and Bagmati had the highest contribution of farmland to urban or forest. As dominant and important areas for these kinds of LCC, the two provinces also had more population than the others with frequent human activities. Most provinces of Nepal are located in multiple ecological regions homogeneously, except for Madhesh. All of Madhesh was in the Terai region and its contribution of population was far above LCC. From CN in Figure 9, the combined land change and population of the seven interest districts played a significant role in the national ones with good representativeness of the country. Populous districts were more likely to have an active change and interaction between urban area, farmland and forest. With the value of 19%, CN in urban change was

higher than forest or farmland change, which also showed the impact of population on urban expansion.

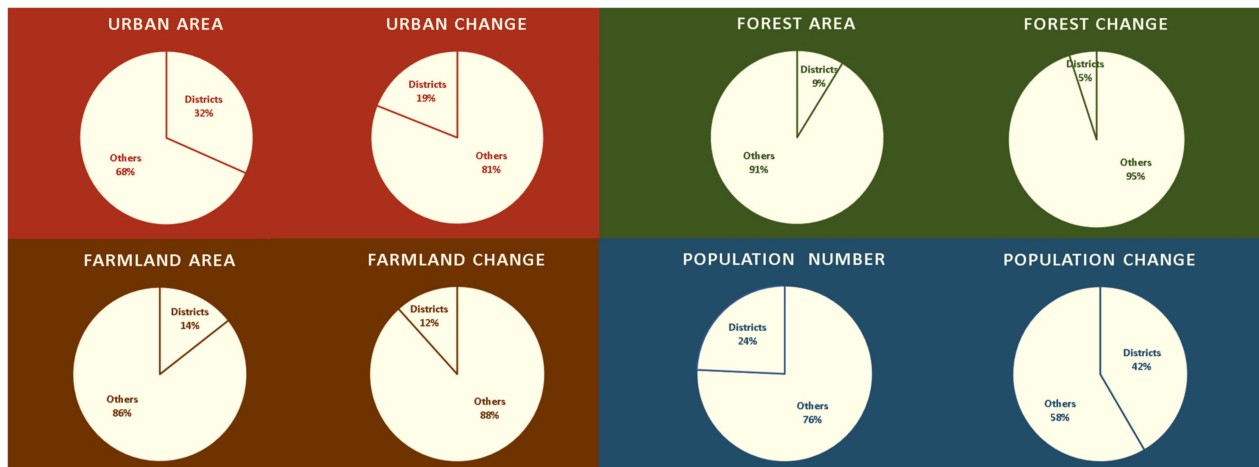

| District (%) | ● | ● | ● | ● | ● | ● |
|---|---|---|---|---|---|---|
| Morang | 4.38 | 21.57 | 2.82 | 4.06 | 21.29 | 23.07 |
| Dhanusa | 3.10 | 17.56 | 4.73 | 12.61 | 13.97 | 14.25 |
| Kathmandu | 3.50 | 17.72 | 0.61 | 1.78 | 31.54 | 33.16 |
| Kaski | 8.36 | 21.49 | 7.68 | 9.42 | 20.47 | 24.18 |
| Rupandehi | 1.12 | 12.15 | 1.94 | 2.43 | 19.56 | 21.84 |
| Surkhet | 21.59 | 17.91 | 7.64 | 29.88 | 22.34 | 24.65 |
| Kailali | 9.88 | 48.58 | 8.86 | 10.55 | 30.39 | 33.61 |

| Province (%) | ● | ● | ● | ● | ● | ● |
|---|---|---|---|---|---|---|
| Province No.1 | 24.31 | 17.74 | 18.00 | 21.50 | 17.12 | 17.03 |
| Madhesh | 0.46 | 7.22 | 0.32 | 2.04 | 20.40 | 20.99 |
| Bagmati | 21.20 | 29.49 | 15.36 | 27.43 | 20.87 | 20.84 |
| Gandaki | 15.43 | 4.75 | 12.18 | 15.52 | 9.07 | 8.49 |
| Lumbini | 13.15 | 23.44 | 10.52 | 14.76 | 16.98 | 17.55 |
| Karnali | 11.48 | 9.82 | 25.47 | 8.31 | 5.93 | 5.81 |
| Sudurpashchim | 13.97 | 7.54 | 18.14 | 10.43 | 9.63 | 9.29 |

**Figure 8.** District's contribution to province (DP) and province's contribution to nation (PN) from 2016 to 2019. The contribution represents the proportion of specific land change area or population at the first, smaller scale to the one at the second, larger scale.

**Figure 9.** Seven districts' combined contribution to nation (CN) from 2016 to 2019. It shows that the total area and change of the seven districts' urban, farmland and forest areas occupied a part of the national change.

From Table 3, the relationship of conversions between farmland, forest and grassland was close. For instance, correlation between farmland to forest and forest to farmland was above 0.90 in both DP and PN, and the one between farmland to grassland and forest to farmland was 0.70 in DP and 0.58 in PN. Contribution of farmland to urban area had a positive relevance of 0.75 with farmland to forest in PN, and the correlation between forest to farmland and farmland to urban was 0.53 in PN. The results showed that some land changes were not isolated processes and had correlation with each other. Additionally, population was more relevant to farmland to urban areas than other LCCs, which reached

the value of 0.62 for PN and 0.54 for DP from the 2011 census. It demonstrated that the population factor was more sensitive to urban expansion.

**Table 3.** Correlation matrix of different LCC types' contributions between multiple scales. DP means the correlation in district to province and PN means the correlation in province to nation.

| Correlation in DP and PN | Forest to Farmland | | Farmland to Urban | | Farmland to Grassland | | Farmland to Forest | |
|---|---|---|---|---|---|---|---|---|
| | DP | PN | DP | PN | DP | PN | DP | PN |
| Forest to farmland | 1.00 | 1.00 | | | | | | |
| Farmland to urban | 0.21 | 0.53 | 1.00 | 1.00 | | | | |
| Farmland to grassland | 0.70 | 0.58 | 0.61 | 0.08 | 1.00 | 1.00 | | |
| Farmland to forest | 0.91 | 0.91 | 0.05 | 0.75 | 0.66 | 0.32 | 1.00 | 1.00 |
| 2011 Population | 0.15 | 0.02 | 0.54 | 0.62 | −0.01 | −0.63 | −0.18 | 0.31 |
| 2021 Population | 0.20 | −0.02 | 0.57 | 0.62 | 0.09 | −0.65 | −0.16 | 0.28 |

From Table 4, the values of RN in various regions showed larger internal diversity than DP and PN. The Hill region had the largest contribution in forest to farmland and farmland to grassland or forest, while urban expansion was prominent in Terai. The Mountain region had the smallest effects on farmland to urban and forest areas. The correlation of LCC between farmland, forest and grassland was high, which reached up to 1.00 between farmland to forest and forest to farmland. This was consistent with DP and PN. The correlation between population and farmland to urban areas in RN was also high.

**Table 4.** Region's contribution to nation (RN) from 2016 to 2019.

| Contribution RN (%) | Forest to Farmland | Farmland to Urban | Farmland to Grassland | Farmland to Forest |
|---|---|---|---|---|
| Terai | 12 | 50 | 7 | 13 |
| Hill | 80 | 45 | 74 | 76 |
| Mountain | 8 | 5 | 19 | 11 |
| **Correlation in RN** | **Forest to farmland** | **Farmland to urban** | **Farmland to grassland** | **Farmland to forest** |
| Forest to farmland | 1.00 | | | |
| Farmland to urban | 0.45 | 1.00 | | |
| Farmland to grassland | 0.98 | 0.25 | 1.00 | |
| Farmland to forest | 1.00 | 0.43 | 0.98 | 1.00 |
| 2011 population | 0.40 | 1.00 | 0.20 | 0.38 |
| 2021 population | 0.29 | 0.98 | 0.08 | 0.27 |

### 4.4. LCC Simulation Result

A CA-Markov simulation was utilized to predict land cover of 2019 based on the data of 2016 and 2017. The transition probability calculated by Markov chain represents the probability that one land cover class will change to other class. Note that the areas of the four land cover classes differ widely in the seven districts. Farmland dominates in Morang, Dhanusa and Rupandehi, while Kaski, Kailali and Surkhet have more forests. Dense urban areas are concentrated in Kathmandu. With no consideration of spatial information, the Markov probability is influenced by the base area of land cover, which means it can only reflect the possibility of conversion rather than total area of change. As shown in Table 5, urban area was relatively persistent, more than 84% of which in most districts was likely to remain the same in the next year. Forest was as stable as urban with about an 83%

unchanged probability, although it changed into farmland greatly in some districts, such as Kathmandu and Morang. Compared to urban and forest areas, farmland and grassland were more active in LCC, with lower unchanged probabilities. Farmland had a potential to turn into forest and be abandoned, especially in the Hill district with a probability of more than 10%. Probability of transition from farmland to urban area reached up to 16.26% in Kathmandu, which was also high in Rupandehi. Kathmandu and Rupandehi were more populous than other districts with a higher possibility of urban expansion. There was also a large mutual transformation between forest and grassland.

**Table 5.** Transition probability matrix of seven districts calculated by Markov chain (unit: %). Urban, farmland, forest and grassland are chosen from nine land cover classes of ICIMOD data.

| Morang | Forest | Urban | Farmland | Grassland | Dhanusa | Forest | Urban | Farmland | Grassland |
|---|---|---|---|---|---|---|---|---|---|
| Forest | 83.54 | 0.06 | 11.19 | 4.94 | Forest | 83.81 | 0.00 | 0.53 | 15.57 |
| Urban | 0.10 | 84.03 | 5.07 | 7.27 | Urban | 0.01 | 84.00 | 11.41 | 0.01 |
| Farmland | 6.00 | 3.33 | 83.58 | 0.87 | Farmland | 1.87 | 2.59 | 83.56 | 0.08 |
| Grassland | 8.70 | 6.89 | 3.20 | 73.47 | Grassland | 18.48 | 0.03 | 0.16 | 79.46 |
| Kathmandu | Forest | Urban | Farmland | Grassland | Rupandehi | Forest | Urban | Farmland | Grassland |
| Forest | 82.94 | 0.58 | 14.63 | 1.85 | Forest | 84.51 | 0.04 | 7.36 | 7.04 |
| Urban | 0.98 | 84.73 | 10.13 | 3.77 | Urban | 0.01 | 84.23 | 10.21 | 0.01 |
| Farmland | 7.38 | 16.26 | 76.16 | 0.19 | Farmland | 4.54 | 5.31 | 83.81 | 0.42 |
| Grassland | 6.66 | 17.01 | 2.23 | 74.09 | Grassland | 34.04 | 0.57 | 7.03 | 56.06 |
| Kaski | Forest | Urban | Farmland | Grassland | Surkhet | Forest | Urban | Farmland | Grassland |
| Forest | 84.00 | 0.01 | 8.60 | 6.96 | Forest | 84.10 | 0.02 | 9.77 | 5.97 |
| Urban | 0.11 | 82.57 | 8.50 | 0.62 | Urban | 0.09 | 84.59 | 4.64 | 3.03 |
| Farmland | 16.25 | 2.04 | 79.88 | 1.31 | Farmland | 14.07 | 1.85 | 81.50 | 2.11 |
| Grassland | 2.08 | 0.12 | 0.59 | 70.06 | Grassland | 14.19 | 0.02 | 9.32 | 75.42 |
| Kailali | Forest | Urban | Farmland | Grassland | | | | | |
| Forest | 84.36 | 0.01 | 7.08 | 6.91 | | | | | |
| Urban | 0.04 | 84.60 | 6.45 | 0.94 | | | | | |
| Farmland | 6.85 | 3.02 | 83.18 | 1.50 | | | | | |
| Grassland | 15.73 | 0.06 | 3.77 | 76.32 | | | | | |

Based on the analysis part, farmland and urban areas were influenced by different driving factors, which could be utilized as suitability maps for land suitability evaluation as shown in Figure 4. Ecological divisions of the Terai and Hill regions were closely related to slope and elevation. The Hill districts, with a complex topography, had more farmland loss than the Terai districts. Related to the growth of farmland, areas with a lower slope and elevation tended to have a suitable climate, abundant labor force and better accessibility to agricultural infrastructure or markets. Surkhet had obvious areas converted from forest to farmland around the river, and distance to river was associated with the supply of water resources for farmland. Therefore, slope, elevation and distance to river were gathered as suitability maps for farmland. Population and transport condition were crucial to the speed and size of urban expansion from the result of multi-scale spatiotemporal pattern analysis. Correlation between population and urban expansion was higher than other LCCs for DP, PN and RN. In addition, flat topography can influence the distribution of urban concentrated areas and bring conveniences to expansion, and urban growth in the Terai districts was a good example. To sum up, maps of population, distance to road and slope were selected as suitability maps for urban area. The seven districts that lay in Nepal from east to west and from Terai to Hill region had large diversities in the driving factors. For instance, slope was higher in the Hill districts than Terai districts, and the appropriate

range of slope for urban area was also different from the one for farmland. Therefore, the adaptive land suitability rule was employed with the above driving factors.

In Figure 10, the prediction result was compared with land cover data of 2019 to demonstrate the performance of the simulation part in the MSPAS model, i.e., the adaptive land suitability rule with the CA-Markov method. In addition, the relationship between the analysis part and the simulation part was also explored. Urban area was projected to expand and occupy farmland from the previous extent in 2016, which was consistent with the distribution of urban area in 2019. The prediction of urban expansion was close to the actual change and particularly obvious in some urban concentrated areas, such as the existing urban area of Kathmandu, the center of Surkhet, the southwest of Kailali and Morang. However, the expansion was slightly excessive in some areas, e.g., a scattered small-scale urban area in Morang. Compared with dense urban areas, these scattered areas had lower expansion speed and a different change pattern. From Table 6, most districts obtained PA and UA of more than 85% as a demonstration of proper performance. Owing to the overestimated area of farmland, UA was relatively lower in Kathmandu, Kaski and Surkhet than in other districts. These districts belonged to the Hill region with a tendency of farmland loss by spatiotemporal analysis of the MSPAS model. The transition probability of Markov reflected the magnitude of LCC quantitively, but neglected spatial information, and the CA-Markov simulation added spatial information to the dynamic process. Conversely, there also existed spatial effects of local differences and irregular change which could influence the performance of simulation. The above results implied that dynamic simulation and spatiotemporal analysis in the MSPAS model had relevance to each other, rather than being two separate processes. From the result of the analysis part, Hill districts had a higher possibility of farmland loss than others, and then farmland area in these districts was over predicted in the simulation part. Scattered and small urban areas had a lower expansion speed, which was overestimated by the Markov transition probability of the whole area in the simulation part. The analysis part can provide references for a better simulation in the MSPAS model.

**Table 6.** Producer's accuracy (PA) and User's accuracy (UA) of farmland simulated by the CA-Markov model with the adaptive land suitability rule.

| Districts (%) | PA | UA | Districts | PA | UA |
|---|---|---|---|---|---|
| Morang | 87.91 | 97.50 | Dhanusa | 85.19 | 99.44 |
| Kathmandu | 86.82 | 78.63 | Rupandehi | 86.71 | 99.44 |
| Kaski | 89.44 | 78.57 | Surkhet | 89.60 | 79.56 |
| Kailali | 90.64 | 91.95 | | | |

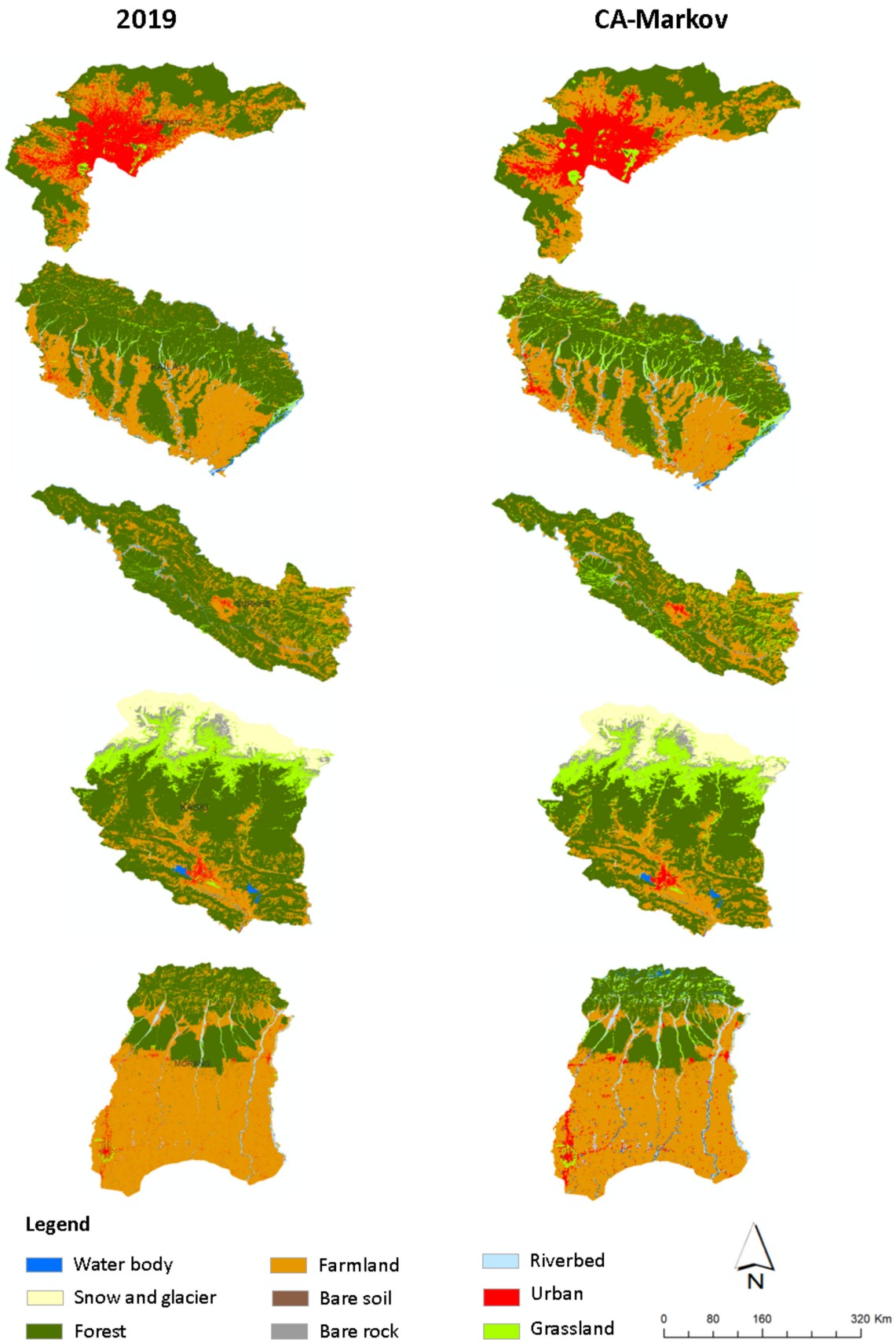

**Figure 10.** Land cover data of 2019 and the prediction of the CA-Markov simulation for 2019. The results of Kathmandu, Kailali, Surkhet, Kaski and Morang are shown here.

## 5. Discussion

### 5.1. Performance and Theoretical Significance of the MSPAS Model

Taking Nepal as a case study, the research proposed a MSPAS model to explore the spatiotemporal pattern of LCC based on multi-scale analysis criteria and simulating the dynamic process. In the analysis part, the result showed that the LCC pattern had consistency and characteristics between different scales. Urban area expanded while farmland decreased in both the nation and seven interest districts, and conversion from farmland to urban or forest areas accounted for a large proportion of LCC. There were also internal characteristics at a single scale. The interest districts were divided into the Terai districts, Hill districts and Kathmandu, which had large differences in spatiotemporal pattern and driving factors. Dominant transition, size, fragmentation degree and distribution of the changed area also varied in these districts. Urban expansion at the expense of farmland was relatively intense in Kathmandu [50] and the Terai districts, while farmland abandonment was prominent in the Hill districts [51].

In multi-scale analysis, contribution and correlation of LCC or population in PN, DP, RN and CN were explored quantitatively. The result showed that multi-scale contribution can measure the significance of components, first at the small scale to the second, large scale for specific LCC dynamic processes, such as a district to its corresponding province or a province to the whole country. Surkhet was crucial to its province in farmland to forest from DP, and Bagmati contributed the most to the whole country in farmland to forest or urban areas from PN, which should be taken seriously in farmland protection and food security. With a large proportion of forest to farmland in PN, Province No.1 needed to keep the balance between ecological protection and agricultural development. In addition, the factor of population was related to LCC, especially for urban expansion. Seven populous interest districts played a greater role in urban change than farmland or forest change from CN. The correlation can quantify the relationship between different land change types or population at multiple scales and deeply imply the rule that LCC is influenced by driving forces. For instance, correlations between contributions of forest to farmland and farmland to forest were all higher than 0.9 in RN, DP and PN, which means that mutual conversion between farmland and forest reached a balance between different scales. There is a possibility that the two variations were both determined by the same driving force, such us population and inner migration, and the LCC types influenced by human activity were likely to have close correlation with each other. Population was more relevant with farmland to urban areas than other LCCs because of its strong driving force and sensitivity to urban expansion.

There existed diversities in multi-scale patterns. Most provinces of Nepal were across two or three ecological regions homogeneously with a relatively small disparity in ecological environment. However, seven interest districts had a large disparity in physical and socio-economic driving factors. For instance, Kathmandu was dominant in urban expansion, and farmland to forest was particularly severe in Surkhet. Multi-scale correlations were affected by the inner disparities, which could explain that correlation between LCC and population was mostly higher in PN than DP. Different countries had diverse characteristics of administrative and regional divisions, which also influenced the multi-scale LCC pattern and calculation of contribution or correlation. Most provinces in Nepal contained two or three regions, while in other countries each province can be covered by only one ecological or geographic region, such as plain or mountain. LCC patterns of provinces or districts in the different regions of mountain, hill, plain and other categories had specific characteristics. Mountainous area was prone to farmland loss and flat area corresponded to urban expansion. Classifying the provinces or districts based on regional division and studying their patterns in a specific country were worth exploring, such as the classification of the Terai districts, Hill districts and Kathmandu in the research.

There were also consistencies and similarities across all scales of PN, DP, RN and CN. For instance, farmland to forest and forest to farmland had a high correlation in different multi-scale situations, and farmland to urban area was more relevant to population

than other land changes. This meant that land changes were not isolated processes with connection to each other across multiple scales. The patterns of the seven interest districts in various ecological regions were consistent with the RN result. The combined interest districts also played a significant role in the whole country with certain representativeness. By exploring multi-scale patterns, a LCC at a lower scale can be utilized to deduce the situation of one at a larger scale. The relationship between different land changes or population can also be analyzed quantitatively for multi-level land management.

In the simulation part, the adaptive land suitability rule was defined using a CA-Markov model due to the diversities of driving factors in different land covers and areas. The result tested the performance of the simulation method with certain effectiveness, which also showed that the simulation part had relevance to the spatiotemporal analysis part in the MSPAS model. The spatial effect of local difference and irregular change explored from the analysis part was consistent with the result of the simulation part, such as farmland loss in the Hill districts and slower expansion in scattered small urban areas. Spatiotemporal analysis focused on the change that has occurred, and dynamic simulation was designed to predict the change in the future. How to add the knowledge from LCC pattern analysis to the dynamic method for a better simulation is one of the future research goals for this study.

### 5.2. Application of the MSPAS Model for SDGs

As a member committed to SDGs, Nepal took an early lead in launching the national SDG road map. In Nepal's progress assessment report of SDGs from 2016 to 2019, the progress of SDG 2 and SDG 11 remained slow and needs to be highly valued [52]. Development of SDGs and related limitations can be explored in Nepal with good representativeness. National Urban Development Strategy in 2017 proposed to balance regional and national urban systems. The Land Use Act of Nepal in 2019 also emphasized to promote the development of land management at multi-level political and administrative units, i.e., federal, provincial and local levels, which were consistent with multi-scale analysis of the spatiotemporal pattern in this study. The MSPAS model can provide scientific references for multi-level administration and land management in different countries to promote sustainable development of urbanization and agriculture. Policies, planning, relevent targets and indicators were combined with the multi-scale and multi-temporal analysis result by the MSPAS model, which explored the development and lessons for SDGs from the perspective of land area, land change and the impact of driving factors on land suitability.

5.2.1. The MSPAS Model for SDG 11 in Urban Expansion at the Expense of Farmland

SDG 11 focuses on sustainable urbanization and inclusive cities. Urban expansion of Nepal increased rapidly and contributed to 33% of national GDP [53]. According to the 2011 census, 17.1% of the Nepalese population lived in 58 designated urban areas, and this number increased to 40% after the addition of 159 municipalities in 2014–2015 [53]. Based on the analysis result, the pace of urbanization in Nepal developed fast at the expense of farmland loss. Existing research suggested that this change was widespread over the past few decades [35]. The trend lasted into the years from 2016 to 2019 with urban area almost doubling and expanding at both national and district scales in this study. As shown in Figure 11, there was a negative correlation between the change of urban and farmland, which implied that the increase of urban area was closely related to the loss of farmland. It is necessary to make a trade-off between urban expansion and the protection of farmland, corresponding to the balance between developing an urban economy in SDG 11 and ensuring food production in SDG 2. Indicators of Target 11.1 for inclusive urbanization and affordable housing achieved a better score than expected, while Target 11.6 for the reduction of adverse environmental impact obtained worse results [52]. Overall, urban development in Nepal achieved positive results in terms of quantity, but was not stable with some problems such as unplanned urban sprawl, lack of management and environmental pollution.

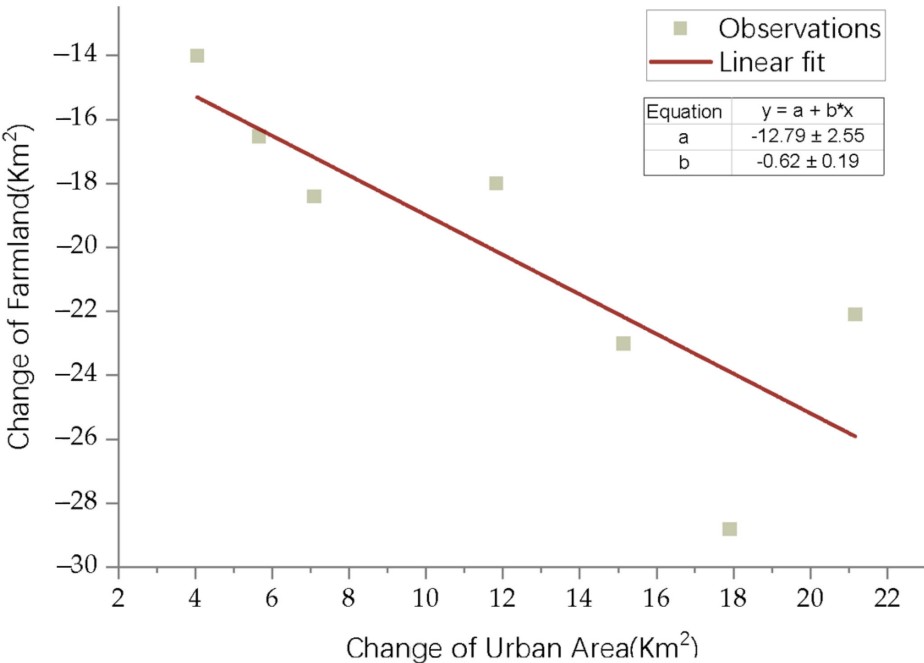

**Figure 11.** Correlation analysis between the change of urban area and farmland. The observations are obtained from the seven interest districts of the study.

Various kinds of driving factors can influence the speed and spatial distribution of the above expansion, including population, transport, topography and previous degree of urbanization [54]. The study found that population was closely related to urban expansion [55]. Some populous districts had large areas converted from farmland to urban area, such as Kathmandu and the Terai districts. As the hub of urbanization with a large population and growth speed shown in Table 7, Kathmandu [56] had a compact and intense expansion around the previous urban centers [57]. Province No.1 occupied the highest percentage of national agricultural and forest GDP, and its populous district of Morang also had the largest area of farmland in all seven districts. However, the problem of urban encroachment on farmland was also obvious in these agricultural areas from the analysis result of the MSPAS model. On the contrary, urban expansion was relatively slight in the Hill districts with a small population and low growth. Dense population and frequent human activities can provide strong driving forces for the interaction between urban, farmland and forest areas. Secondly, convenient transport was [58] relevant to the driving force of neighborhoods, such as access to market centers or infrastructure, and flat topography was suitable for urban expansion. Urban sprawl [33] in the Terai districts mostly presented linear distribution along the road or small-scale fragments widely distributed in the flat areas, while the expansion in the Hill districts was centralized in a small area [59] with steep topography. Transport was prioritized as a major bottleneck for the development of Nepal in SDG 9, and SDG 11 also sought to build safe and sustainable urban transport systems. Nepal's Strategic Road Network expanded the roads to 6979 km in 2017, which could increase the connection between isolated districts and the rest of the area, spur farm production by convenient transport and improve living conditions of poor households in the remote areas.

**Table 7.** Population number of the seven districts in 2011 and 2021 from the census of Nepal. National GDP represents the proportion of agriculture and forestry GDP in the province to that of Nepal, and provincial GDP means the proportion of agriculture and forestry GDP to the province's total GDP.

| Study District | Population | | Growth in 10 Years | Province | GDP of Agriculture and Forestry (%) | |
|---|---|---|---|---|---|---|
| | 2011 | 2021 | | | Nation | Province |
| Morang | 965,370 | 1,147,186 | 181,816 | Province No.1 | 21.53 | 36.37 |
| Dhanusa | 754,777 | 873,274 | 118,497 | Madhesh | 19.00 | 37.90 |
| Kathmandu | 1,744,240 | 2,017,532 | 273,292 | Bagmati | 17.09 | 12.96 |
| Kaski | 492,098 | 599,504 | 107,406 | Gandaki | 9.95 | 29.91 |
| Rupandehi | 880,196 | 1,118,975 | 238,779 | Lumbini | 17.31 | 30.18 |
| Surkhet | 350,804 | 417,776 | 66,972 | Karnali | 5.39 | 32.85 |
| Kailali | 775,709 | 911,155 | 135,446 | Sudurpashchim | 9.73 | 37.98 |

### 5.2.2. The MSPAS Model for SDG 2 in Agriculture and Forestry

SDG 2 aims to develop sustainable agriculture, double agricultural productivity and achieve food security. For SDG 2, ensuring the area of cultivated land is a crucial task. In Nepal, about two thirds of the national population were engaged in agriculture, and farmland accounted for 24% of the whole area in 2019, contributing to 26% of GDP [60]. This study has found that farmland area had a decline of 3% from 2016 to 2019. Although the steady loss was related to urban expansion, farmland abandonment also led to the decrease of arable land [61]. The phenomenon was particularly significant and widespread in the Hill districts [36], which exhibited scattered small-scale patches of the forest and hill areas. Different driving factors affected the change of farmland and forest [62]. From Table 7, Gandaki and Karnali had low percentages of national GDP with great likelihood of a poor agricultural condition. Their corresponding interest districts of Kaski and Surkhet had a smaller area and size of farmland, but suffered more intense farmland abandonment than others. In mountainous areas with steep terrain [63], poor transport conditions and sparse population, farmlands are fragmented and far away from water resources or markets, which means it is hard to maintain the normal agriculture activities and obtain high production. Additionally, high altitude also leads to complex topography, unsuitable climate and a short growing season with many restrictions on agriculture. Therefore, these farmlands have a high possibility of abandonment.

Some abandoned farmland turned into forest or grassland and promoted forest growth indirectly. The increase of forest also benefited greatly from the community level forest management system and practice [64,65], which promoted a sustainable terrestrial ecosystem and obtained encouraging progress for SDG 15. Nepal has built a network of protected areas and it consisted of 12 national parks, 1 wildlife reserve, 1 hunting reserve, 6 conservation areas and 13 buffer zones, covering 23.39% of the whole country. ICIMOD and other institutions also made contributions to the forest protection.

Some acts were developed by the Nepalese government to support agriculture into a sustainable high growth development. National Land Use Policy in 2015 gave priority to the protection of farmland by commercial farming and land consolidation. Then the Land Use Act was endorsed in 2019 to regulate the management through land classification based on topography and suitability, which attached great significance to the farmland. However, most measures focused on land use management nationally and neglected differences in a local level, such as geographical condition and population [66]. The policy of land management should strengthen the interaction between different scales, corresponding to the analysis of multi-scale spatiotemporal patterns in the study. The local administration needs more capacity to develop land use plans with an active participation of local people, communities and households. For food security, the target of per capita food grain production has been achieved in SDG 2, but there still exists urgent problems

for improvement. Target 2.1 for global food security developed slightly, and 7.8% of the population suffered food insecurity. Target 2.4 for agricultural production and productivity was also not encouraging [52].

### 5.2.3. The MSPAS Model for Migration and Natural Disasters

Moreover, urban expansion and farmland abandonment are not two completely isolated processes which have interactions with each other by effects such as inner rural–urban or mountain–plain migration. There is a steady trend that people migrate from mountainous areas to plain areas for agriculture activities, or move from rural areas to urban areas. The census in 2011 revealed that there were 1.5 million internal migrations in Nepal, 61% of which immigrated to the Terai region [67]. Notably, people from rural areas accounted for 77% of urban migrations [68]. Urban areas have better education opportunities, living conditions and services than rural areas. Many young people and hill farmers become inner migrants to diversify their sources of income. They may move to a new place with better agricultural conditions and continue to farm, or give up and choose another job in an urban area. The original farmland gradually turns into forest. This can explain the high correlation of mutual conversion between farmland and forest, which is also consistent with the phenomenon that area from farmland to forest was always more than the one from forest to farmland in Figure 6. Natural disasters such as earthquake [69], flood [70] and landslide [71] bring dramatic damage to the agriculture and urban development. With high rainfall, steep terrain and the possibility of active earthquake, Nepal is vulnerable to natural disasters. The earthquake of 2015 broke down the buildings, roads and infrastructure, and affected agricultural activities for over a year [72]. Aiming at the problem, Target 2.4 of SDG 2 is devoted to building a sustainable food production system to improve the adaptation to disasters, and Target 11.5 of SDG 11 focuses on the reduction of loss caused by disaster.

## 6. Conclusions

The study proposed a MSPAS model to analyze the multi-scale spatiotemporal patterns of LCC and simulate the dynamic process by the adaptive land suitability rule with driving factors. The model took Nepal as a case study and focused on LCC of four specific land covers from 2016 to 2019, i.e., farmland, urban, forest and grassland. Contributions of DP, PN, RN and CN were calculated, and correlations between different LCCs or population were quantified. The results showed that urban area nearly doubled in Nepal, especially in Kathmandu and the Terai districts, while farmland had a steady loss, and the area decreased by 3% in these years. By population information from the census, the most populous district in each province was chosen as an interest district. As the hub of urbanization, Kathmandu expanded its urban area to the nearby farmland. Urban encroachment on farmland mostly occurred along the roads in the Terai districts. Farmland abandonment was prominent and intense in the Hill districts with steep topography and sparse population, most of which presented scattered small-scale patches widely distributed in the hills and forests. The performance of the MSPAS model showed that there were characteristics and consistencies at different scales, and population was closely related to the LCC of urban expansion. The impacts of physical and socio-economic driving factors on LCC were also analyzed. The simulation result of the MSPAS model was consistent with the spatiotemporal analysis result, which means that the LCC pattern from the analysis part can be utilized for a better simulation. Finally, the model was applied to SDGs with political factors, which can provide references for sustainable development of urban and agricultural areas in a specific country. Future study will collect different kinds of data and driving factors to analyze the spatiotemporal process. Exploring criteria to further quantify the impacts of various factors is worth studying, which can utilize the quantitative and spatial correlation between driving factors and LCC. The multi-scale situations can be expanded to more patterns, such as province to region and district to region. Finally, how to utilize the knowledge from the analysis part to better simulate the dynamic process is also a significant point of

study. The application of the MSPAS model for SDGs can also be systematized and linked to quantitative index and targets of SDGs [73].

**Author Contributions:** Conceptualization, W.J., X.M. and X.G.; methodology, W.J. and X.M.; formal analysis, W.J. and X.M.; data curation, W.J., P.L., Z.Z., X.Z. and H.Z.; writing—original draft preparation, W.J.; writing—review and editing, X.M. and J.Y. (Jian Yan); Visualization, W.J.; project administration, X.M., J.Y. (Jian Yang) and W.Z.; funding acquisition, X.M. All authors have read and agreed to the published version of the manuscript.

**Funding:** This research was funded by the National Key R&D Program of China (2020YFE0200700 and 2019YFE0127300).

**Data Availability Statement:** Not applicable.

**Acknowledgments:** The authors are very grateful to the support of International Centre for Integrated Mountain Development (ICIMOD).

**Conflicts of Interest:** The authors declare no conflict of interest.

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
