# Peer review of "Multi-Scale Spatiotemporal Pattern Analysis and Simulation (MSPAS) Model with Driving Factors for Land Cover Change and Sustainable Development Goals: A Case Study of Nepal"

_remotesensing, doi:10.3390/rs14246295_

Round 1
Reviewer 1 Report (Previous Reviewer 1)
After careful consideration and deliberation by the authors of this manuscript, it is clear that the document underwent thorough revision and was subject to numerous modifications. It is interesting to note that the way in which the results, the introduction, and the discussion section are presented have all seen substantial improvements. Consequently, looking at it from this perspective, I believe it is acceptable, and I suggest that it be published.
Author Response
We sincerely appreciate your willingness to allow us to revise the manuscript. Thanks a lot again for your precious time and previous kind suggestions, which help us make the manuscript better!
Reviewer 2 Report (Previous Reviewer 2)
The revised paper has been greatly improved. The model was clearly described with sufficient detail and logic. I think the paper can be published in the Remote Sensing after a minor revision.
The section 3. Study Area and Materials could be moved to as section 2, and the section 2. MSPAS Model with Driving Factors could be Section 3 as you used the map and data in the method introduction.
Figure 6. is the data used for analysis but not results, it should be in section of materials.
The land cover data of 2016 in Figure 10 has been shown in Figure, it should be not shown again.
Author Response
Thank you very much for your suggestion! Please see the attachment.

Reviewer 3 Report (Previous Reviewer 3)
This research has done a great job on multi-scale spatiotemporal pattern analysis in Nepal. The results revealed that the urban area nearly doubled only in 4 years after 2015, and was mainly transferred by farmland. The authors have done thoughtful and considerable revisions. Some points are not very clear and I suggest improving.
l Section 2.2, how did you exactly decide the driving factors and the impact of each factor? Directly naming them seems not credible. Furthermore, how you calculate the specific impact of each factor in each area, which seems to need to be put into the model.
l Figure 2, the main point the figure wants to express is OK, but the figure is not clear and missing of legend.
l In the 4.2.1 quantitative transition analysis part, using the absolute number of changing areas to compare may not reflect the actual intensive differences of different regions, owing to the different original areas of each region.
l Figure 5, both "Farmland to forest" and "Forest to Farmland" are obvious in many regions, the reason might be interesting if you could provide more discussion.
l Section 4.4, since you already have the actual land cover of 2019, why built a CA-Markov model to only simulate a virtual land cover in 2019?
Author Response
Thank you very much for your suggestion! Please see the attachment.

This manuscript is a resubmission of an earlier submission. The following is a list of the peer review reports and author responses from that submission.
Round 1
Reviewer 1 Report
see the attached file.

Reviewer 2 Report
This paper studied the spatial-temporal pattern of Nepal’s land cover change from 2016 to 2019 at multiple scales. The authors used adaptive land suitability rules and CA-Markov model to predict land cover of 2019. The paper was well organized with plenty of analysis. However, the paper lacks of scientific significance and clear research objectives. The article also lacks certain innovation in method and research content. So I do not think it can be published in Remote Sensing considering its current version. Some comments for the author.
According to my understanding, the land suitability evaluation should be serve for CA Markov model simulation? Why is it a parallel relationship in the flowchart? I am also confused why the authors have to simulate the land cover 2019 as it is in a so short interval.
The author should give a more thoroughly analysis about the relationships and differences among different spatial scales?
Result: The results section can be more concise and the text should be avoided that can be directly read from the tables and figures.
Discussion: I'm not familiar with SDG, but I don't understand how the author's research results can be related to SDG? In particular, it mainly focuses on qualitative discussions without in-depth analysis and related data. The discussion did not correspond to the results, and there was no targeted discussion around the results.
Reviewer 3 Report
This manuscript analysed the spatial-temporal change of land use land cover in Nepal from 2016 to 2019, the work and the results are meaningful considering the timing of Federal Republic establishment in 2015 and the rapid urban expansion. This work would be interesting to readers focusing on relevant fields. Here are some suggestions:
In figure 1, it is better to use a DEM or one land cover map as a base map to get a direct impression of the topography of the study area and also good for the hill land abandonment discussion.
In the results section 3.4, that would be more convictive to compare the LULC and simulation results in 2019 point by point spatially to provide a consolidated accuracy of the CA-Markov model.
The driving factors analysis of Land Use/Cover Change in Nepal is one of the main objects of your research, it should be pointed out clearly. Besides, please explain how these driving factors have been considered in the CA-Markov model.
The land Use/Cover Change model is used to simulate future scenarios and thus to support land-use decision-making for example on SDGs, however, how is the CA-Markov model work in this research?
